# Influence of Patient-Reported Outcome Measures by Surgical Versus Conservative Management in Adult Ankle Fractures: A Systematic Review and Meta-Analysis

**DOI:** 10.3390/medicina59061152

**Published:** 2023-06-15

**Authors:** Hui Peng, Xiao-Bo Guo, Jin-Min Zhao

**Affiliations:** 1Department of Trauma Orthopedics and Hand Surgery, The First Affiliated Hospital of Guangxi Medical University, Nanning 530021, China; 2Minzu Hospital of Guangxi Zhuang Autonomous Region, Nanning 530000, China

**Keywords:** adult ankle fractures, surgical treatment, conservative treatment, joint function and physical health

## Abstract

*Background and Objective*: This meta-analysis was performed to compare the effectiveness of surgical treatment and conservative treatment in adult ankle fractures. *Methods*: Pubmed, Embase, and Cochrane-Library databases were searched to retrieve prospective randomized-controlled studies that compared the efficacy of surgical treatment and conservative treatment in adult ankle fractures. The meta package in R language was used to organize and analyze the obtained data. *Results*: A total of eight studies involving 2081 patients was considered eligible, including 1029 patients receiving surgical treatment and 1052 receiving conservative treatment. This systematic review and meta-analysis was prospectively registered on PROSPERO, and the registration number is CRD42018520164. Olerud and Molander ankle-fracture scores (OMAS) and the health survey 12-item Short-Form (SF-12) were used as main outcome indicators, and the follow-up outcomes were grouped according to the follow-up time. Meta-analysis results showed significantly higher OMAS scores in patients receiving surgical treatment than those with conservative treatment at six months (MD = 1.50, 95% CI: 1.07; 1.93) and over 24 months (MD = 3.10, 95% CI: 2.46; 3.74), while this statistical significance was absent at 12–24 months (MD = 0.08, 95% CI: −5.80; 5.96). At six months and 12 months after treatment, patients receiving surgical treatment exhibited significantly higher SF12-physical results than those receiving conservative treatment (MD = 2.40, 95% CI: 1.89; 2.91). The MD of SF12-mental data at six months after meta-analysis was −0.81 (95% CI: −1.22; 0.39), and the MD of SF12-mental data at 12+ months was −0.81 (95% CI: −1.22; 0.39). There was no significant difference in SF12-mental results between the two treatment methods after six months, but after 12 months, the SF12-mental results of patients receiving surgical treatment were significantly lower than those of conservative treatment. *Conclusions*: In the treatment of adult ankle fractures, surgical treatment is more efficacious than conservative treatment in improving early and long-term joint function and physical health of patients, but it is associated with long-term adverse mental health.

## 1. Introduction

Ankle fractures are a common type of fracture, accounting for approximately one in ten fractures [1]. In recent years, with the aging of the population and increasing traffic accidents, the prevalence of ankle fractures is on the rise [2]. The ankle joint is composed of a synovial mortise and tenon involving the articular surface of the tibia (ankle and talus). It works in conjunction with the subtalar joint and acts as an improved hinge that allows plantar flexion, and bending, sliding, and rolling of the dorsal side. The ankle is also bounded by three lateral ligaments and a strong medial delta ligament. Therefore, this is a complex area with many potential sites of injury [3,4]. Improper treatment of ankle fractures, dislocation, or ligament injury is associated with poor joint- function recovery. Ankle fractures are intra-articular fractures and require complete anatomical reduction to avoid posttraumatic arthritis [5]. Surgical treatment and conservative treatment are the main treatment modalities for ankle fractures, used to restore the normal structure and function of the ankle joint. 

Surgical treatment usually involves incision, repositioning of the fracture site, and internal fixation of the fracture using synthetic bone materials. In conservative management, manual reduction is followed by casting or braces to stabilize the ankle. Surgical and conservative treatments of ankle fractures are aimed at restoring the anatomy and stability of the ankle joint. Immobilization is considered the optimal procedure for fracture healing to minimize the risk of malunion at the fracture site. However, it also results in reduced overall activity and function [6]. For ankle fractures with displacement, surgical treatment is the preferred treatment [7]. However, research showed that non-surgical treatment combined with regular review can also produce comparable functional recovery to surgery [4,8,9]. A study based on US Commercial and Medicare Supplemental databases showed that ankle fractures affect 0.14% of the population, with 23.4% of fractures requiring surgery. In addition, surgery had no significant effect on functional recovery or complications [10]. Otherwise, non-surgical treatment can significantly reduce the risk of surgery-related adverse events such as venous thromboembolism, infection, failure of internal fixation, and revision [11]. However, conservative treatment prolongs postoperative fixation and increases the risk of ankle stiffness, weakness, swelling, and residual pain.

At present, the optimal treatment for ankle fractures is unknown and the question remains controversial [12]. Clinically, surgical treatment mostly achieves internal fixation via plates and screws, while non-surgical treatment mostly uses manual-reduction plaster casts and external splint fixation. However, the efficacy of the two treatment methods is still controversial [13]. To this end, it is imperative to gather the latest evidence from randomized controlled trials to provide a basis for the choice of treatment for patients with ankle fractures. 

Accordingly, this systematic review and meta-analysis was performed to compare surgical treatment and conservative treatment, so as to provide a reference for the choice of treatment approaches.

## 2. Materials and Methods

This study was guided by the Preferred Reporting Items for Systematic Reviews and Meta-Analyses (PRISMA) guidelines [14]. This systematic review and meta-analysis were prospectively registered on PROSPERO.

### 2.1. Search Strategy and Selection Criteria

We searched Pubmed, Embase, and Cochrane-Library from the dated of their inception to April 11, 2022. We used the search terms (((“fractures, bone”[MeSH Terms])) AND ((“ankle”[MeSH Terms] OR “ankle joint”[MeSH Terms]))), in English and Chinese. We reviewed the reference lists of all included articles and all pertinent review articles to identify articles the database searches may have missed.

### 2.2. Inclusion and Exclusion Criteria of Literature

#### 2.2.1. Inclusion Criteria

(1) Type of study: randomized controlled clinical trial (RCT); (2) Subject: Ankle fracture diagnosed by imaging [15]; (3) Surgical treatment and conservative treatment, follow-up duration ≥ 6 months; (4) Research indicators: Olerud and Molander ankle fracture score (OMAS) [16] and one or more items of the 12-item Short-Form (SF-12) [17] Health Survey; (5) The research design is scientific and standardized, with clear groupings and interventions, follow-up data.

#### 2.2.2. Exclusion Criteria

(1) Unextractable OMAS, SF-12, and other related outcome indicators; (2) combination with other treatment methods or secondary surgery; (3) participants being younger than 18 years; (4) patients undergoing secondary surgery; (5) the number of patients being greater than 10, otherwise it would be more like a case report.

### 2.3. Quality Assessment

Two reviewers independently screened the search results, retrieved full-text articles, checked inclusion criteria, and eliminated duplicate studies from three levels of article title, abstract and full text, and then confirmed the included articles of this study. The two independent reviewers evaluated the quality of the included literature using the Cochrane Collaboration Risk Bias Assessment Tool [18], and a consensus was reached by consulting a third reviewer in the case of discrepancies.

### 2.4. Data Extraction

Two reviewers independently extracted data and entered them in electronic forms including the first author’s name, year of publication, number of patients, and outcome measures. The primary outcome of interest was OMAS, and the secondary outcome was SF-12. OMAS was completed according to patients’ self-assessment, and the content mainly included nine items, namely, pain, joint stiffness, joint swelling, going up stairs, running, jumping, squatting, walking, and working ability, with a full score of 100 points. This scoring system is currently the most widely used ankle-fracture efficacy scoring system. The SF-12 is a 12-item questionnaire used to assess generic health outcomes from the patient’s perspective. Generic patient-reported outcome measures such as the SF-12 assess general health and well-being, including the impact of illnesses on a broad range of functional domains.

### 2.5. Statistical Methods

We conducted meta-analyses using the R software meta package. The OMAS and SF-12 data extraction included the total number of cases (n), mean and standard deviation (sd), and the analysis results were expressed as (Mean difference, MD, 95% CI). An MD greater than 0 indicated that the results more favored the surgical group. We estimated the heterogeneity via *I*^2^ test (i.e., *I*^2^ > 0% or *p*-value < 0.1 indicated heterogeneity, and a random-effects model of analysis was used; *I*^2^ = 0% or *p*-value > 0.1 indicated the absence of significant heterogeneity, and a fixed-effects model of analysis was used). Funnel plots were used to describe publication bias, and Egger’s test was used to test for funnel plot asymmetry.

## 3. Results

### 3.1. Results of the Literature Search and Intervention Studies

Our search yielded 548 citations, which were initially screened on the abstract level for eligibility. After excluding case reports, abstracts, and reviews, 505 pieces of literature were excluded, and 43 studies were coarsely included. After reading the full text, 8 studies including 2081 patients were deemed eligible (1029 cases of surgical treatment and 1052 cases of conservative treatment) [19,20,21,22,23,24,25,26]. The search flow chart is shown in Figure 1. Risk assessment of bias in the included studies is shown in Figure 2, but blinding of participants was unavailable due to study limitations. The descriptive details of the included trials are provided in Table 1.

### 3.2. OMAS Analysis

#### 3.2.1. OMAS Analysis after 6 Months of Treatment

As shown in Figure 3A, 4 studies, including 1646 patients, had OMAS results after 6 months of treatment. The heterogeneity test showed that *I*^2^ = 0%, and the fixed-effects model was used for analysis.

#### 3.2.2. OMAS Analysis after 12–24 Months of Treatment

As shown in Figure 3B, 3 studies analyzed OMAS results after 12 to 24 months of treatment, involving 186 patients. The heterogeneity test showed that *I*^2^ = 0%, and the fixed-effects model was used for analysis. After meta-analysis, the MD of OMAS score at 12–24 months was 0.08 (95% CI: −5.80; 5.96), and there was no significant difference in OMAS score between surgical treatment and conservative treatment at 12–24 months.

#### 3.2.3. OMAS Analysis after 24 Months of Treatment

As shown in Figure 3C, 4 studies included OMAS data after 24+ months of treatment, involving 617 patients. The heterogeneity test showed that *I*^2^ = 15%, and the fixed-effects model was used for analysis. After meta-analysis, the MD of 24+ months OMAS score was 3.10 (95% CI: 2.46; 3.74), and the OMAS score of surgical treatment was significantly higher than that of conservative treatment, and the difference was statistically significant.

### 3.3. SF-12 Analysis

#### 3.3.1. SF12-Physical Analysis

A total of 6 studies analyzed the SF12-physical results of patients, with SF12-physical results being within 6 months in 4 studies and 12+ months data in 2 studies. As shown in Figure 4A, *I*^2^ = 34% in SF12-physical at 6 months, and a fixed-effect model was used for analysis. The MD of SF12-physical data at 6 months after meta-analysis was 1.53 (95% CI: 1.20; 1.87). As shown in Figure 4B, *I*^2^ = 46% in SF12-physical data at 12+ months, and a fixed effect model was used for analysis, the MD of SF12-physical data at 12+ months after meta-analysis was 2.40 (95% CI: 1.89; 2.91). At 6 months and 12 months after treatment, patients receiving surgical treatment exhibited significantly higher SF12-physical results than those receiving conservative treatment (MD = 2.40, 95% CI: 1.89; 2.91).

#### 3.3.2. SF12-Mental Analysis

A total of 6 studies included treatment data of SF12-mental, including 4 studies with 6-month results and 2 studies with 12+-month results. As shown in Figure 4C, *I*^2^ = 0% in SF12-mental at 6 months, and a fixed-effect model was used for analysis. The MD of SF12-mental data at 6 months after meta-analysis was −0.11 (95% CI: −0.52; 0.30); as shown in Figure 4D, *I*^2^ = 0% in SF12-mental at 12+ months, and a fixed effect model was used for analysis. The MD of SF12-mental data at 12+ months after meta-analysis was −0.81 (95% CI: −1.22; 0.0.39). There was no significant difference in SF12-mental between the two treatments at 6 months after treatment, but after 12 months, the SF12-mental results of patients receiving surgical treatment were significantly lower than those of conservative treatment.

### 3.4. Publication Bias

The publication bias of different research papers is shown in Figure 5. The publication funnel plots of all analyzed papers are symmetrically distributed, suggesting a low risk of publication bias in this study.

## 4. Discussion

Ankle fractures are one of the most common fractures in adults, and treatment approaches should be appropriately selected based on correct classification of the injury and soft tissue damage [24]. The goal of ankle-fracture treatment is to restore weight-bearing capacity in the ankle without pain [25]. A total of 2081 patients were included in the present meta-analysis, including 1029 cases treated by surgery and 1052 cases treated by conservative treatment, with a time span from 2001 to 2022. The results of this study showed that meta-analysis results showed significantly higher OMAS scores in patients receiving surgical treatment than those with conservative treatment at 6 months (MD = 1.50, 95% CI: 1.07; 1.93) and over 24 months (MD = 3.10, 95% CI: 2.46; 3.74), while this statistical significance was absent at 12–24 months (MD = 0.08, 95% CI: −5.80; 5.96). In this study, the SF-12 physical score of the surgical treatment was significantly higher than that of the conservative treatment; but after 12 months, the SF 12-mental score of the surgical treatment was significantly lower than that of the conservative treatment, and the SF12-mental results of patients receiving surgical treatment was significantly lower than those receiving conservative treatment.

OMAS is the primary index for evaluating ankle-joint function [26]. The current study shows that although there is no significant difference in mid-term functional recovery between the two approaches, surgical treatment has potential benefits for long-term functional recovery. The major principle of ankle-fracture management is to restore the anatomical position and joint consistency, ensure stability, and reduce long-term complications. Stable fractures with slightly-displaced or no fragments can be treated conservatively [27]. Weber A fractures do not require cast immobilization, but ankle orthoses can be used to immobilize the ankle for early functional recovery [28]. For other types of fractures, external fixation should be performed within six weeks after the injury to limit the motion of the ankle joint and the bearing of weight. Conservative treatment should pay attention to the combination of fracture-fragment displacement and ankle acupoint widening, and X-ray follow-up is recommended at 4, 7, 11 and 30 days after injury [15].

Some research suggested that surgical treatment in elderly patients is unnecessary. For patients with a high surgical risk, older age, and multiple underlying diseases, conservative treatment without anatomic reduction is recommended [23]. Ankle fractures are categorized as intra-articular fractures, for which the purpose of surgery is to reconstruct the anatomy and protect the damaged ligament structure for early postoperative functional treatment of the joint. However, poor joint alignment is more common than direct injury to the articular surfaces of the tibia, talus, or fibula [26]. In addition, soft tissue damage is another factor that affects the timing of surgery, as periarticular swelling may increase the risk of necrosis and infection and compromise surgical outcomes. One study reported a three-year outcome of cast fixation versus conventional surgery for unstable ankle fractures in older people and found the same outcome of ankle function at six months [22]. Under conservative treatment, immobilization of the ankle joint with an orthosis resulted in relief of joint pain and improvement to weight-bearing capacity of the joint, which may be a similar result of conservative and surgical treatment in terms of functional recovery [28]. However, in long-term postoperative recovery, the functional recovery of surgical treatment is significantly superior to that of conservative treatment. SF-12 is a self-reported measure that assesses the impact of health on an individual’s daily life and is mainly categorized into physical and mental aspects. Quality of life is a key aspect to evaluate the effect of treatment, and the effect of physical health in this study is consistent with the results of OMAS. However, surgical treatment has a certain impact on long-term mental health. As age increases, basic health status declines, and the history of surgery inevitably poses psychological pressure on patients, but a specific database of mental health is lacking in the study. In addition, Bauer et al., reported no significant difference in long-term pain between surgery and conservative treatment [22].

There were some limitations associated with the small number of papers included in this study, plus the great variation in the time span and treatment methods. There are great differences between the surgical and conservative treatments of ankle fractures in different operators, and it is necessary to choose the best treatment according to the individual situations of patients, such as age, gender, and physical ability. In the future, it is necessary to explore the factors influencing the effect of surgery and conservative treatment to provide patients with more accurate guidance.

## 5. Conclusions

In the treatment of adult ankle fractures, surgical treatment is more efficacious than conservative treatment in improving early and long-term joint function and physical health, but it is associated with long-term adverse mental health.

## Figures and Tables

**Figure 1 medicina-59-01152-f001:**
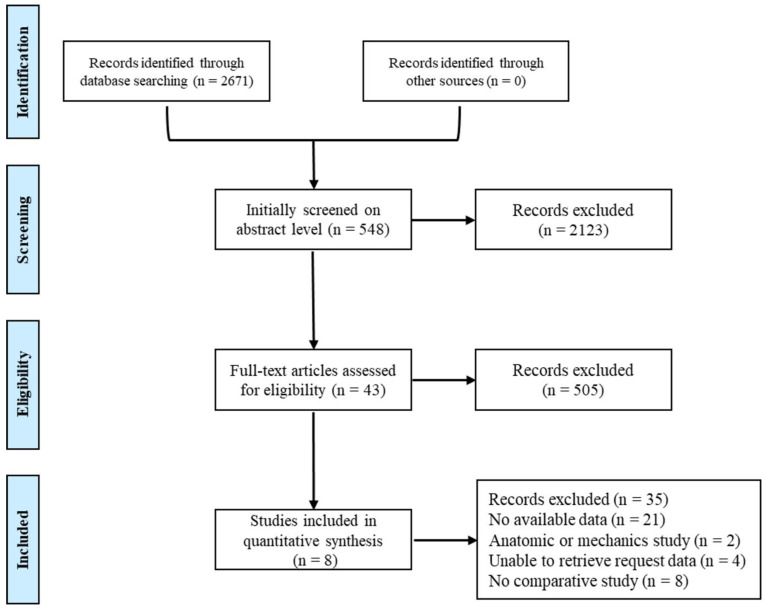
Search flow chart.

**Figure 2 medicina-59-01152-f002:**
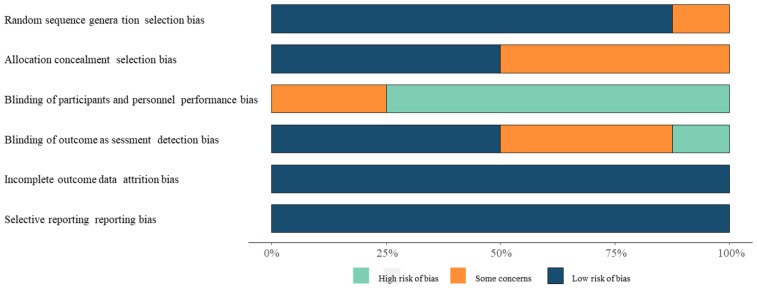
Risk of bias assessment chart.

**Figure 3 medicina-59-01152-f003:**
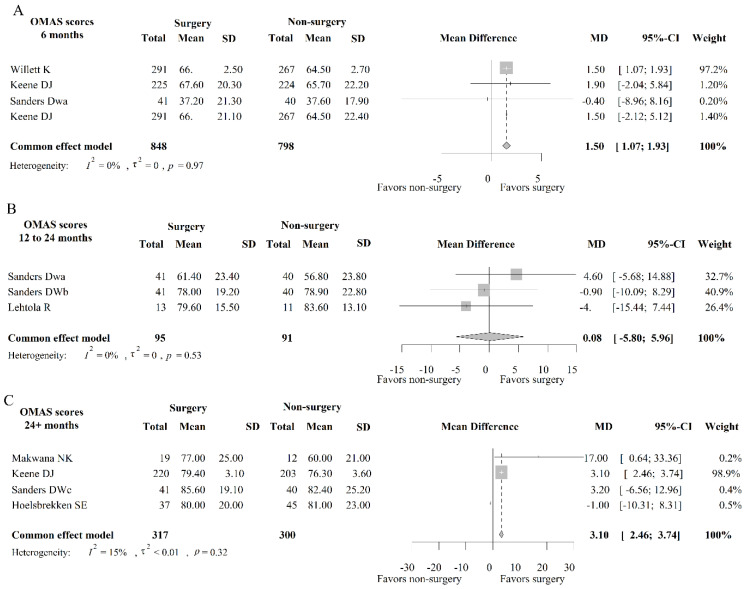
Forest plot of OMAS scores at different times. Note: (**A**) indicates the OMAS scores 6 months post intervention; (**B**) indicates OMAS scores during 12 to 24 months post intervention; (**C**) indicates OMAS scores more than 24 months post intervention.

**Figure 4 medicina-59-01152-f004:**
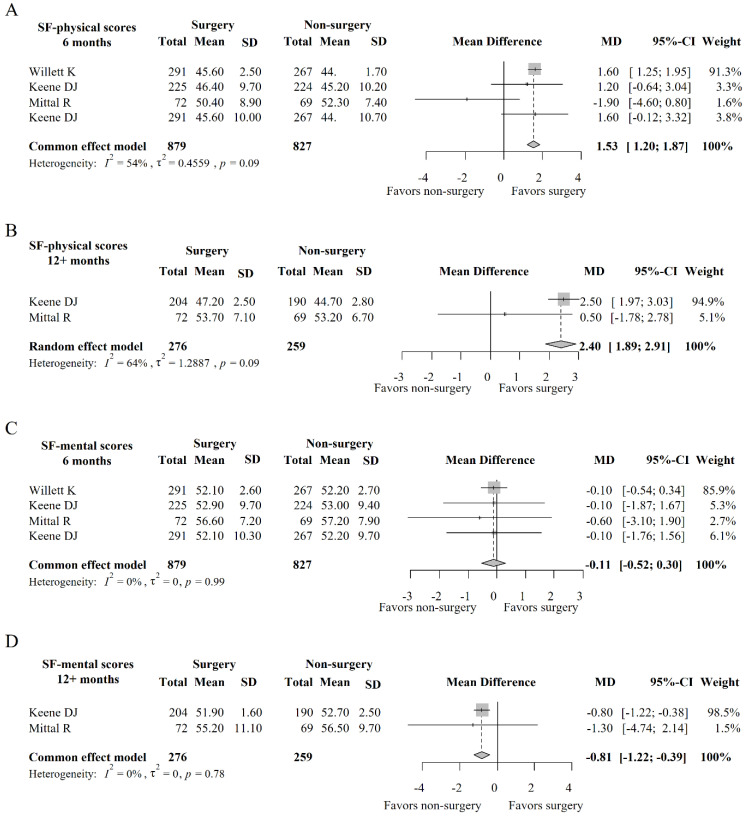
Forest plot of SF-12 scores at different times. Note: (**A**) indicates the SF-12 physical scores 6 months post intervention; (**B**) indicates the SF-12 physical scores 12 months post intervention; (**C**) indicates the SF-12 mental scores 6 months post intervention; (**D**) indicates the SF-12 mental scores 12 months post intervention.

**Figure 5 medicina-59-01152-f005:**
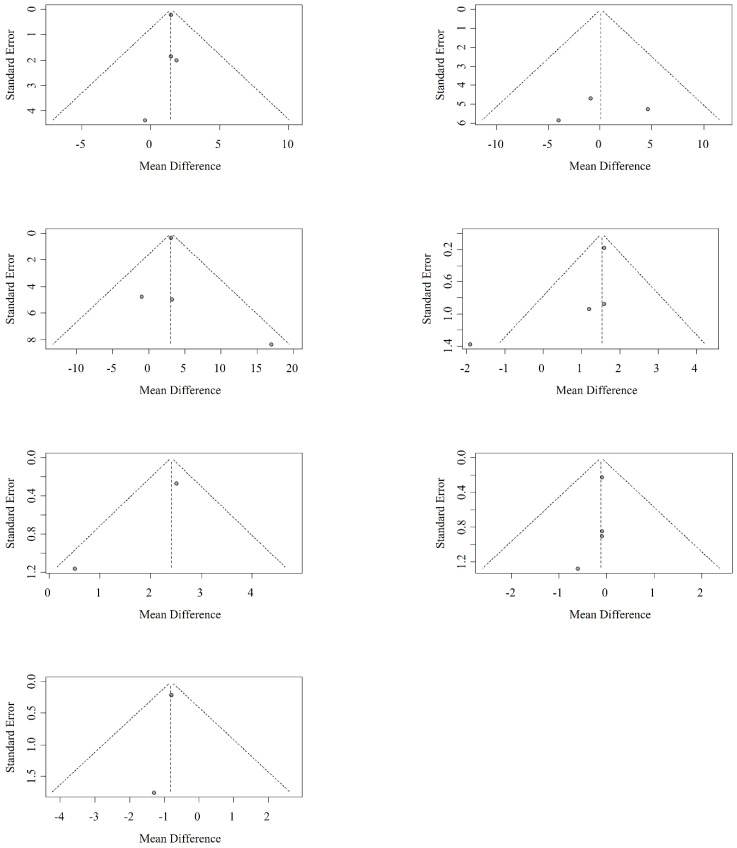
Funnel plots of all papers.

**Table 1 medicina-59-01152-t001:** Descriptive details of the included trials.

Author	Year	Gender (Male/Female)	Age	Surgery	Non-Surgery	Main Indicators
Makwana NK [20]	2001	12/31	66 (57 to 77)	22	21	OMAS
Willett K [21]	2016	160/460	70.2 ± 7.4	309	311	SF-12, OMAS
Keene DJ [22]	2018	119/331	69.6 ± 6.7	226	224	SF-12, OMAS
Mittal R [23]	2017	56/104	38.9 ± 13.3	72	69	SF-12
Sanders DW [24]	2012	41/40	41	41	40	OMAS
Lehtola R [25]	2022	15/9	43.6 ± 12.6	13	11	OMAS
Keene DJ [26]	2016	180/280	69.6 ± 11.2	309	311	SF-12, OMAS
Hoelsbrekken SE [27]	2013	31/51	52.8 ± 15.7	37	45	OMAS

## Data Availability

All data generated or analyzed during this study are included in this published article.

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
