# Peer review of "Influence of Patient-Reported Outcome Measures by Surgical Versus Conservative Management in Adult Ankle Fractures: A Systematic Review and Meta-Analysis"

_medicina, 2023, doi:10.3390/medicina59061152_

Round 1

Reviewer 1 Report (Previous Reviewer 2)

Title

Looking at the title, it's too broad. Didn't weber B investigate only the case where there is no displacement of the talus? Please change the title to be more restrictive.

Introduction

“The latest statistics show that non-surgical treatment combined with regular review

can achieve functional recovery comparable to surgery.”

: What is the reference for this statement? A previous review also pointed out this point, but it was not corrected.

: Please add more references to this background.

“the efficacy of the two treatment methods is still controversial”

: Weber B was investigated only in the method, but it is unreasonable to use this expression in all ankle fractures.

: There is definitely an "op indication", but can we ignore this and do conservative treatment?

I can't agree.

Materials and Methods

: Well organized.

Results

The quality of figure is too poor.

Figure: The figure is blurry and hard to see. Please upload again.

This is also not an improvement compared to before.

Conclusion

“But there is currently insufficient evidence to determine whether surgery or conservative.”

As I mentioned in the introduction, I can't agree with this opinion. A simple literature search comparison cannot arrive at this result.

Discussion

“Despite the small number of literatures included in this study and great variation in the time span, treatment methods, grouping methods, rehabilitation guidance, follow-up time and other aspects, this study also yielded meaningful results.:

: In fact, this is the most important limitation of this study. Please elaborate on this limitation.

“Type A fractures do not require cast imm ~~”

: Are you talking about "weber classification"? Please refer to the weber type correctly.

From the next review, please create a number before the line.

Author Response

Title

Looking at the title, it's too broad. Didn't weber B investigate only the case where there is no displacement of the talus? Please change the title to be more restrictive.

Reply: The title was revised.

Introduction

“The latest statistics show that non-surgical treatment combined with regular review

can achieve functional recovery comparable to surgery.”

: What is the reference for this statement? A previous review also pointed out this point, but it was not corrected. Reply: The refs were added.

: Please add more references to this background. Reply: More related information was added.

“the efficacy of the two treatment methods is still controversial”

: Weber B was investigated only in the method, but it is unreasonable to use this expression in all ankle fractures. Reply: The sentence was modified to limit Weber B to fracture.

: There is definitely an "op indication", but can we ignore this and do conservative treatment?

I can't agree. Reply: The fracture type is limited to Weber B with no talus displacement, the surgery and conservative treatment all are available.

Materials and Methods

: Well organized.

Results

The quality of figure is too poor.

Figure: The figure is blurry and hard to see. Please upload again.

This is also not an improvement compared to before.

Reply: HD vector pictures are provided as accessories.

Conclusion

“But there is currently insufficient evidence to determine whether surgery or conservative.”

As I mentioned in the introduction, I can't agree with this opinion. A simple literature search comparison cannot arrive at this result.

Reply: In the treatment of adult ankle fractures, surgical treatment is more efficacious than conservative treatment in improving early and long-term joint function and physical health, but it is associated with long-term adverse mental health. This is the main conclusion of this study. Some description was revised to make it clearer.

Discussion

“Despite the small number of literatures included in this study and great variation in the time span, treatment methods, grouping methods, rehabilitation guidance, follow-up time and other aspects, this study also yielded meaningful results.:

: In fact, this is the most important limitation of this study. Please elaborate on this limitation.

Reply: This is the limitation of this study, which has a certain impact on the results of the meta-analysis. It was discussed at the end of the Discussion.

“Type A fractures do not require cast imm ~~”

: Are you talking about "weber classification"? Please refer to the weber type correctly.

Reply: Yes, revised.

From the next review, please create a number before the line.

Reviewer 2 Report (New Reviewer)

Titles: I suggest modifying your titles based on the aim of your research focus: patient-reported outcome measures 

Introduction

1. Please clarify the background knowledge (what we have known and what is still uncertain), and then point out the aim of this MA/SR wound to investigate.

Methods.

1. Please point out the reviewers who work on the literature selection and who make the decision when disagreements meet. Inter-rater reliability?

2. "(2) Less than 10 patients were included in a single group?" How do you decide the number of participants in a group is enough to be included? Why 10, but not 5 or 20?

3. Please clarify this study's primary outcome and secondary outcome. Please give a short description of how to interpretation of these OMAS ad SF-12.

4. Please provide the positivity of MD indicating flavoring surgery or nonsurgical treatment.

Results.

1. Figure 1&2: The quality needs to be improved. Difficult to read the contents.

2. Table 1. Are any other PROMs reported in your included studies? The follow-up period for each study.

3. Figure 3&4. Please make the direction of "Favors surgery" or "Favor nonsurgery" at the bottom of the Forest plot. 

Discussion

1. "As the age increases, the basic health status declines, and the history of surgery inevitably poses psychological pressure on the patient, but the specific database of mental health is lacking in the study."  What is the MCID of the SF-12 mental score? How do you know the MD -0.81 is clinically relevant?

2. Since you considered long-term adverse mental health as the major finding in this study, you need to figure out the reason, for example, worry of a syndesmotic screw broken? Do this long-term adverse mental health in the surgical group happen in other fracture, e.g., distal radius fracture?

Author Response

Comments and Suggestions for Authors

Titles: I suggest modifying your titles based on the aim of your research focus: patient-reported outcome measures 

Reply: The title was revised.

Introduction

  1. Please clarify the background knowledge (what we have known and what is still uncertain), and then point out the aim of this MA/SR wound to investigate.

Reply: The background knowledge was enriched in the introduction.

Methods.

  1. Please point out the reviewers who work on the literature selection and who make the decision when disagreements meet. Inter-rater reliability?

Reply: “Two independent reviewers evaluated the quality of the included literature by Cochrane Collaboration Risk Bias Assessment Tool, and if consensus was not reached a third reviewer was involved.”

  1. "(2) Less than 10 patients were included in a single group?" How do you decide the number of participants in a group is enough to be included? Why 10, but not 5 or 20?

Reply: It is generally considered that when the sample size is less than 10 people, it should be considered case report rather than an RCT. However, this view is unfounded, and we have removed this clause from the exclusion criteria, and after re-evaluation, it has no impact on the results.

  1. Please clarify this study's primary outcome and secondary outcome. Please give a short description of how to interpretation of these OMAS ad SF-12.

Reply: “Among them, the primary outcomes of interest was OMAS and the secondary outcome was SF-12.”

  1. Please provide the positivity of MD indicating flavoring surgery or nonsurgical treatment.

Reply: Added.

Results.

  1. Figure 1&2: The quality needs to be improved. Difficult to read the contents. Reply: Revised.
  2. Table 1. Are any other PROMs reported in your included studies? The follow-up period for each study. Reply: Study-relevant data were added, and only a few studies defined follow-up time.
  3. Figure 3&4. Please make the direction of "Favors surgery" or "Favor nonsurgery" at the bottom of the Forest plot. Reply: Added.

Discussion

  1. "As the age increases, the basic health status declines, and the history of surgery inevitably poses psychological pressure on the patient, but the specific database of mental health is lacking in the study."  What is the MCID of the SF-12 mental score? How do you know the MD -0.81 is clinically relevant? Reply: The results of the study confirm the statistical correlation, and there is no further discussion and evidence for the MCID of the SF-12 mental score.
  2. Since you considered long-term adverse mental health as the major finding in this study, you need to figure out the reason, for example, worry of a syndesmotic screw broken? Do this long-term adverse mental health in the surgical group happen in other fracture, e.g., distal radius fracture? Reply: The specific database of mental health is lacking in the study, so the exact reason cannot be explained.

Reviewer 3 Report (New Reviewer)

The manuscript deals with a very common disease and, therefore, it is essential to acquire as much knowledge as possible to implement the best type of treatment. This is the aim of this review which is overall well structured, however, some additional insight could improve the article and make it worthy of publication.

In particular, the introduction is somewhat lacking in describing the disease as a whole (prevalence, trauma mechanisms, clinical aspects etc.)

Materials and methods are correctly described and the authors have made the study reproducible.

The section of results, with the aid of the tables, make undestranding fluent and intuitive; the conclusions are in agreement with the results.

In the limitations of the study, the authors should consider and add the heterogenicity of the surgical treatment and the fact that different types of surgical treatment give different results in terms of function and quality of life.

The article needs to a general language revision.

Author Response

Comments and Suggestions for Authors

The manuscript deals with a very common disease and, therefore, it is essential to acquire as much knowledge as possible to implement the best type of treatment. This is the aim of this review which is overall well structured, however, some additional insight could improve the article and make it worthy of publication.

In particular, the introduction is somewhat lacking in describing the disease as a whole (prevalence, trauma mechanisms, clinical aspects etc.)

Reply: The related information was added.

Materials and methods are correctly described and the authors have made the study reproducible.

The section of results, with the aid of the tables, make undestranding fluent and intuitive; the conclusions are in agreement with the results.

In the limitations of the study, the authors should consider and add the heterogenicity of the surgical treatment and the fact that different types of surgical treatment give different results in terms of function and quality of life.

The article needs to a general language revision.

Reply: Polished

Round 2

Reviewer 1 Report (Previous Reviewer 2)

In conclusion, mention the research on weber B mentioned above.

Author Response

Thank you for your comments, we have revised it as requested.

Reviewer 2 Report (New Reviewer)

1. The authors did not revise the manuscript according to my previous suggestion. For example, the who (author's initials) did the literature review and the registry number of this study.

2.  I cannot tell how to make why the conclusion that "Weber A" needs surgery rather than conservative treatment based on your inclusion criteria and results.

Author Response

  1. The authors did not revise the manuscript according to my previous suggestion. For example, the who (author's initials) did the literature review and the registry number of this study.

Reply: It‘s claimed in the “Authors' contributions” that “Hui Peng and Xiaobo Guo performed the literature review”. The registry number was provided in the abstract and Materials and Methods.

  1. I cannot tell how to make why the conclusion that "Weber A" needs surgery rather than conservative treatment based on your inclusion criteria and results.

Reply: I’m so sorry that it should be Weber B in the conclusion.

This manuscript is a resubmission of an earlier submission. The following is a list of the peer review reports and author responses from that submission.

Round 1

Reviewer 1 Report

In this study, the authors used a meta-analysis to compare the efficacy and adverse events of surgical and conservative treatment for ankle fractures, and the results showed that surgical treatment could play a good role in long-term joint function recovery in patients with ankle fractures. However, this study still has some problems.

1.        In the results section of the abstract, the authors need to indicate whether the combined effect size is OR or RR, which may refer to the following format: OR/RR=? , 95% CI=? -? , P=?.

2.        Could the authors provide a complete MeSH terms?

3.        Could the authors provide a search flow chart with reasons for inclusion and exclusion?

4.        In Table 1, the feature extraction of the research included in this study is not perfect, and basic information such as the country of the study and the ratio of male to female of the research object should be supplemented.

5.        Some formatting in the article needs to be modified, such as repeated periods (Line 18), word spelling, repeated/missing Spaces between words (Line 71), percentage sign not added (Line 103), etc.

6.        Figure 1-3 and Figure 4-7 can be merged respectively.

7.        The comment section of the Figure was too concise. The author should add an introduction to the figures.

Reviewer 2 Report

Introduction

The latest statistics show that non-surgical treatment combined with regular review

can achieve functional recovery comparable to surgery.”

: This reference alone is weak. Please add more references to this background.

The choice of clinical

treatment mostly depends on the preference of the surgeon or the needs of the patient,

which lacks medical evidence-based support.”

: Of course, I agree with the opinions of the authors. But what do the authors think of "op indications"? Reading this sentence, might the reader have the same thoughts as above? Please comment on this.1. Please clearly mark the distinction between "arrow" and "thick arrow" in Fig.1. There is no difference in the picture, so it is hard to tell them apart.

Materials and Methods

: Well organized.

Results

Figure : The figure is blurry and hard to see. Please upload again.

Conclusion

Please compose the Conclusion item separately.

Discussion

Despite the small numberof literatures included in this study and great variation in the time span, treatment methods,grouping methods, rehabilitation guidance, follow-up time and other aspects, this

study also yielded meaningful results.:

: In fact, this is the most important limitation of this study. Please elaborate on this limitation.

“In addition, for patients with high surgical risk,~~”

; The content of the discussion is too general. Please add more information about what the authors are claiming

Reviewer 3 Report

I believe this is an important topic that there is very little published research. I value the effort of the authors but I have doubts about some aspects related to the quality of the work. It would be advisable to follow the PRISMA statement to perform the systematic review.

Specific comments

Indicate in the list of authors, who is the corresponding author with an *.

Put in the title also that it is a systematic review.

Abstrac

Methods:

.... Pubmed, EMBase, and Cochrane-Library databases were searched to retrieve prospective randomized controlled studies comparing surgical treatment and conservative treatment in adult ankle fractures. Add adult’s whit ankle fractures.

Results

...A total of 8 studies including 2081 patients were deemed eligible (1029 cases of surgical treatment and 1052 cases of conservative treatment). Olerud and Molander ankle fracture score (Olerud-Molander Ankle Score, OMAS). Eliminate Olerud-Molander Ankle Score because it is repeated.

The authors do not specify the inclusion and exclusion criteria of the review and the limitations of the review should be stated.

Introduction

The introduction is scarce and does not provide the necessary information. For example, Specific data on the prevalence of fractures could also be provided. In addition, it would be possible to provide information on the main advantages and disadvantages of each treatment.

In addition, it is important to end this section with the objective well established and well identified. The aim of this systematic review and meta-analysis was ......

Methodology

- I do not know if the authors were guided by the Preferred Reporting Items for Systematic Reviews and Meta-Analyses (PRISMA) guidelines. If so, they should mention and reference it.

- When the authors mention the search strategy, the complete search strategies for all databases, registries and websites should be presented, including the filters and limits used. This information should be named in the text as well as I invite you to make a supplementary table with the complete search strategies in each database.

I think that this section should be improved since the way the search strategy is represented is not reproducible.

- In the section on inclusion criteria in point 4, do not put etc, but name the whole list. The research design is scientific and standardized, with clear groupings and interventions, follow-up data, etc..

- Type of study: Randomized Comparison clinical Trial (RCT); the correct is randomized controlled clinical trial.

- Some of the exclusion criteria listed by the authors are the opposite of the inclusion criteria. The exclusion criteria are not the opposite of the inclusion criteria.

- Why was there no language restriction?

Quality assessment

- Why did the authors choose the NEWCASTLE - OTTAWA scale to assess the methodological quality of RCTs? Do the authors believe it is the most appropriate scale? In addition, the authors in this section do not mention how the result of this scale is interpreted. On the other hand, the scale they passed is not referenced.

- In this section you comment that ... Two reviewers examined the search results, retrieved the full-text articles, checked the inclusion criteria, and eliminated duplicate literature on the basis of three levels: article title, abstract, and full text. These two reviewers worked independently on the review?

Data extraction

- Two reviewers independently extracted data and entered them in electronic forms including first author name, year of publication, number of patients, outcome measures, duration of follow-up, and other results. what other results? Wouldn't it be interesting to also extract data on the type of study, intervention, whether or not there was blinding, analysis?

- Why did the authors choose the articles whose response variables were the OMAS and SF-12? It is not argued in the text if they are validated scales, if they are widely used in this population, what is assessed with them?

Results

- In section 3.1 Results of the literature search and intervention studies, we consider that the whole process of article selection should be more specific, clarifying the reasons why some studies were eliminated; were there duplicate articles, were any studies incorporated after reviewing the bibliography of the included studies, ... On the other hand, it would be helpful for readers if a flow chart of the whole process of article selection could be included in the text.

- Why were the authors finally left with only 8 articles?

- I believe that Table 1 should provide more information such as the age of the participants, the surgical (surgical technique) and conservative intervention (what the treatment consisted of), the duration of the study, and it would also be useful for readers to know when the outcome variables were evaluated (at 6, 12 or 24 months), the results of the methodological evaluation ......

- In the statistical analysis section, the authors should explain the meaning of MD the first time it appears in the text, even if it is a common term.

- The images in Forest plots are blurred.

- During the text, in some parts of the text, the authors write 6 months and in others they put 6-months: it is advisable to unify the way of writing in the text. In this sense also in some sections there is a space between MD and 95IC% and in other parts of the text there is no space at all.

- In the results section. Specifically in section 3.3.2. SF12-mental analysis in the second line it says month. I think they are missing an s.

Discussion

- I think that in this section the authors should contrast their results more with the literature. Other studies, previous reviews on this subject? On the other hand, there are several statements that are not referenced.

- In this section it is also important to talk about the limitations of the study, as well as some recommendations for future research.

- Regarding the number of references, I believe that 25 are too few considering that 8 studies are included.

Reviewer 4 Report

Analysis by paper partitions:

1 - Introduction: must be reformed in the content and in the writing of the general part review the syntax of the topic

2- Discussion : deepen in the discussion the various therapies that act in the regulation of osteoclastogenesis and osteoblastogenesis in the problems of fractures, use the following papers (optional) : PMID: 32102398 ; PMID: 29516238 ; PMID: 29924137

  • 3 - Check the bibliographic entries throughout the text, some of which are non-compliant.

4 - Review the English grammar and in particular the applied scientific English: in particular, verbal tenses and syntax in the discussion.